# A Pilot Study Investigating the Role of Gender in the Intergenerational Relationships between Gene Expression, Chronic Pain, and Adverse Childhood Experiences in a Clinical Sample of Youth with Chronic Pain

Jennaya Christensen [1], Jaimie K. Beveridge [2], Melinda Wang [3,4], Serena L. Orr [5], Melanie Noel [2,3,4] and Richelle Mychasiuk [1,2,3,4,*]

1    Department of Neuroscience, Central Clinical School, Monash University, Melbourne, VIC 3004, Australia; Jennaya.christensen@monash.edu
2    Department of Psychology, University of Calgary, Calgary, AB T2N 1N4, Canada; jaimie.beveridge@ucalgary.ca (J.K.B.); Melanie.noel@ucalgary.ca (M.N.)
3    Hotchkiss Brain Institute, Calgary, AB T2N 4N1, Canada; Melinda.wang@ucalgary.ca
4    Alberta Children's Hospital Research Institute, Calgary, AB T2N 4N1, Canada
5    Department of Pediatrics, Cumming School of Medicine, University of Calgary, Calgary, AB T2N 4N1, Canada; Serena.Orr@albertahealthservices.ca
*    Correspondence: Richelle.mychasiuk@monash.edu

**Abstract:** Chronic pain is a highly prevalent and costly issue that often emerges during childhood or adolescence and persists into adulthood. Adverse childhood experiences (ACEs) increase risk for several adverse health conditions, including chronic pain. Recent evidence suggests that parental trauma (ACEs, post-traumatic stress disorder (PTSD) symptoms) confers risk of poor health outcomes in their children. Intergenerational relationships between parental trauma and child chronic pain may be mediated by epigenetic mechanisms. A clinical sample of youth with chronic pain and their parents completed psychometrically sound questionnaires assessing ACEs, PTSD symptoms, and chronic pain, and provided a saliva sample. These were used to investigate the intergenerational relationships between four epigenetic biomarkers (COMT, DRD2, GR, and SERT), trauma, and chronic pain. The results indicated that the significant biomarkers were dependent upon the gender of the child, wherein parental ACEs significantly correlated with changes in DRD2 expression in female children and altered COMT expression in the parents of male children. Additionally, the nature of the ACE (maltreatment vs. household dysfunction) was associated with the specific epigenetic changes. There may be different pathways through which parental ACEs confer risk for poor outcomes for males and females, highlighting the importance of child gender in future investigations.

**Keywords:** biomarker; dopamine; epigenetics; trauma; ACEs; PTSD; parents; children; adolescents

## 1. Introduction

Chronic pain, which is defined as continuous or intermittent pain persisting for at least three months [1], is a substantial burden to the individuals affected, their family, and society [2–4]. In particular, 1 in 5 children and adolescents report chronic pain, which engenders an increased risk of chronic pain and mental health disorders into adulthood [5–7]. Important sex differences exist amongst these conditions, wherein chronic pain prevalence is higher in female children and adolescents compared to their male counterparts [6], which is also the case with the prevalence of internalizing mental health disorders (e.g., anxiety, depression, and eating disorders) [8].

Adverse childhood experiences (ACEs) have repeatedly been linked to an increased risk for several adverse health conditions, including chronic pain, in both youth and adults [9–13]. ACEs have been defined as emotional, sexual, or physical abuse, emotional

or physical neglect, and five types of household dysfunction including household substance abuse, a violent home environment, family mental illness, parental incarceration, and parental separation or divorce [9,12]. The relationship between ACEs and chronic illnesses is frequency-dependent, wherein exposure to more ACEs is associated with greater impairments [12–15]. Emerging research suggests that exposure to certain ACEs (i.e., maltreatment vs. household dysfunction) may also be more predictive of poor health outcomes, including chronic pain and PTSD symptomology [16,17].

Various parental factors critically influence the development and maintenance of chronic pain in children. Most notably, it has been established that offspring of parents with chronic pain have an increased risk of developing chronic pain and the associated negative psychological outcomes (e.g., PTSD symptomology) [2,18]. Evidence also suggests that parental ACEs confer risk of poor health outcomes in children [19–26]. Given that parental trauma symptomology has been shown to play a role in pediatric chronic pain [17–19,23,27–30], it may be that parent ACEs also impact chronic pain in children. Recent evidence suggests that rates of ACEs, particularly maltreatment (physical neglect), are high among parents of youth with chronic pain, which may suggest they confer risk for the onset of child pain [19]. However, little is known about this relationship between parent ACEs and pediatric chronic pain, including the possible neurobiological mechanisms that mediate this relationship.

It has been proposed that the intergenerational transmission of chronic pain is influenced by several mechanisms, including (epi)genetics, neurobiological development, family stress, general parenting, family health, and pain-specific social learning [2]. Genetic factors have been shown to contribute to approximately 50% of the risk for the development of chronic pain in adults [2,31]. Additionally, genetics also affect sensory (e.g., antinociceptive receptors and pain thresholds) and psychological factors (e.g., PTSD symptomology), which are frequently associated with chronic pain [2,31–33]. In youth, the development of chronic pain is likely influenced by the interaction between these genetic factors and both the parent's and child's experiences (including ACEs), environment, and attributes [2]. Epigenetics, which refers to persistent changes in gene expression that alter cellular activity and functional outcomes without affecting the primary DNA sequence [34–36], is an important genetic factor to consider in the intergenerational chronic pain transmission. Epigenetic mechanisms respond dynamically to an organism's environment and experiences by producing changes in behavior [34,36,37]. Several studies have linked physically painful experiences to epigenetic changes that sensitize behavioral responses to stimuli and contribute to the persistence of pain in the same individual [38–41]. However, we propose that it is also possible that the experiences of a parent with chronic pain and/or trauma produce epigenetic modifications in the child that may put them at heightened risk for the development and maintenance of chronic pain.

This brief report is the first to investigate the intergenerational relationship between ACEs, gene expression, chronic pain, and PTSD symptomology in a clinical sample of youth with chronic pain. Additionally, this study sought to determine the capability of examining epigenetic markers via saliva, rather than blood. The genes examined in this study, which include catechol-O-methyltransferase (COMT), dopamine receptor D2 (DRD2), glucocorticoid receptor (GR), and serotonin transporter (SERT), were chosen based on their established link to chronic pain and/or ACEs [31,42–45] (see Methods for detailed rationale). We hypothesized that parent ACEs and/or chronic pain would be significantly related to the child's expression levels of the aforementioned genes, which may then be used as objective biomarkers for identifying risk of chronic pain. Moreover, since the prevalence of both chronic pain and internalizing mental health disorders is higher in females, they may be at the highest risk for these intergenerational epigenetic changes.

## 2. Results

Participant characteristics are reported in Table 1 Parents ranged in age from 36 to 64 years old and were predominately female, White/Caucasian, and married or common-law. Approximately half of the parent sample reported having at least a college de-

gree, full-time employment, and an annual household income greater than $90,000 CAD. Youth ranged in age from 10 to 18 years old and were also predominately female and White/Caucasian. Youth reported an average pain duration of 3.5 years (SD = 3.21 years) and an average pain intensity of 5.55 out of 10 (SD = 1.81). The most commonly reported pain locations among youth were muscle and joints, head, and other. Almost half of the youth sample reported pain in more than one location and almost one-third reported having pain more days than not in the past week. Summary statistics for the key self-report measures are displayed in Table 2.

**Table 1.** Participant characteristics.

| Parent Sample (*n* = 41) | *n* | % or M (SD) |
|:---:|:---:|:---:|
| Age, years | - | 45.29 (5.90) |
| Gender | | |
| Female | 38 | 92.7 |
| Male | 3 | 7.3 |
| Race/ethnicity | | |
| White/Caucasian | 35 | 85.4 |
| Biracial/multiracial | 6 | 14.6 |
| Marital status | | |
| Married or common-law | 33 | 80.5 |
| Separated or divorced | 7 | 17.1 |
| Single | 1 | 2.4 |
| Education | | |
| High school or less | 5 | 12.2 |
| Vocational school or some college (no degree) | 14 | 34.1 |
| College or Bachelor's degree | 19 | 46.3 |
| Graduate/Professional school (Master's degree, PhD) | 3 | 7.3 |
| Employment status | | |
| Full-time | 19 | 46.3 |
| Part-time | 13 | 31.7 |
| Not working | 8 | 19.5 |
| Did not answer | 1 | 2.4 |
| Annual household income, CAD | | |
| 0–29,999 | 4 | 9.8 |
| 30,000–59,999 | 5 | 12.2 |
| 60,000–89,999 | 9 | 22.0 |
| >90,000 | 17 | 41.5 |
| Did not answer | 6 | 14.6 |
| **Child Sample (*n* = 86)** | ***n*** | **% or M (SD)** |
| Age, years | - | 14.07 (2.27) |
| Gender | | |
| Female | 64 | 74.4 |
| Male | 22 | 25.6 |
| Race/ethnicity | | |
| White/Caucasian | 70 | 81.4 |
| Biracial/multiracial | 6 | 7.0 |
| Black | 2 | 2.3 |
| South Asian | 2 | 2.3 |
| Aboriginal/Indigenous | 1 | 1.2 |
| Arab/West Asian | 1 | 1.2 |
| Latin American | 1 | 1.2 |
| Other | 3 | 3.5 |
| Pain duration, years | - | 3.50 (3.21) |

**Table 1.** *Cont.*

| | | |
|---|---|---|
| Pain locations | | |
| Muscle and joints | 31 | 36.0 |
| Head | 30 | 34.9 |
| Legs | 11 | 12.8 |
| Stomach | 6 | 7.0 |
| Chest | 3 | 3.5 |
| Other | 26 | 30.2 |
| Two or more locations | 37 | 43.0 |
| Pain frequency | | |
| Not at all | 15 | 17.4 |
| Once per week | 21 | 24.4 |
| 2 to 3 times per week | 22 | 25.6 |
| 4 to 6 times per week | 9 | 10.5 |
| Daily | 18 | 20.9 |
| Did not answer | 1 | 1.2 |
| Pain intensity, out of 10 | - | 5.55 (1.81) |

Note: n = sample size, M = mean, SD = standard deviation).

**Table 2.** Summary statistics for the key self-report measures.

| Variable | M (SD) | Range | *n* |
|---|---|---|---|
| Parent Total ACEs | 2.34 (2.71) | 0–10 | 41 |
| Parent Total Maltreatment | 1.07 (1.52) | 0–5 | 41 |
| Parent Total Household Dysfunction | 1.27 (1.43) | 0–5 | 41 |
| Parent Chronic Pain Status | - | yes/no | 41 |
| Parent PTSD Symptoms | 9.40 (9.45) | 0–80 | 38 |
| Youth Pain Interference | 55.03 (9.34) | 36.7–74 | 83 |
| Youth PTSD Symptoms | 16.24 (18.58) | 0–80 | 80 |

Note: n = sample size, M = mean, SD = standard deviation.

*Relationship between Gene Expression, Chronic Pain, and ACEs*

One-way ANOVAs examining gender differences in gene expression (parent and child) of *COMT* ($F(2128) = 3.652$, $p = 0.029$), *DRD2* ($F(2128) = 3.268$, $p = 0.041$), *GR* ($F(2128) = 5.498$, $p = 0.005$), and *SERT* ($F(2128) = 4.791$, $p = 0.010$) revealed gender differences in expression between males and females; $p < 0.05$. Therefore, distinct gender-based correlational analyses were run for male and female children. The results from the correlational analyses between parental ACEs and epigenetic targets can be found in Tables 3–6, for female and male children, respectively, while the results from the correlational analyses between parental ACEs and the pain/psychological measures can be found in Tables 5 and 6, for female and male children, respectively. Briefly, for female children, total parental ACEs, and specifically, total maltreatment, was negatively correlated with child expression of DRD2. Although not reaching significance, there was a trend between parental report of total household dysfunction and child expression of SERT. Interestingly, for female children, parental ACEs, specifically total maltreatment, was also positively correlated with parental chronic pain status. Conversely, for male children, parental ACEs, and specifically, total household dysfunction, was negatively correlated with parent expression of COMT. Although not significant, there was a trend towards a significant correlation between total maltreatment and parent expression of DRD2 and GR. Finally, for male children, total parental ACEs and total maltreatment were significantly positively correlated with parental PTSD symptoms.

**Table 3.** Results from the correlational analyses between parental ACE measures and epigenetic targets for female offspring.

| Parental ACE Measure | Epigenetic Target | r | p |
|---|---|---|---|
| Total ACEs | Parent *COMT* | −0.069 | 0.728 |
| | Parent *DRD2* | −0.091 | 0.644 |
| | Parent *GR* | −0.209 | 0.287 |
| | Parent *SERT* | −0.151 | 0.443 |
| | Youth *COMT* | 0.045 | 0.725 |
| | <span style="color:red">Youth *DRD2*</span> | <span style="color:red">−0.264</span> | <span style="color:red">0.037</span> |
| | Youth *GR* | −0.088 | 0.495 |
| | Youth *SERT* | −0.189 | 0.138 |
| Total Maltreatment | Parent *COMT* | −0.170 | 0.388 |
| | Parent DRD2 | −0.120 | 0.543 |
| | Parent GR | −0.217 | 0.267 |
| | Parent *SERT* | −0.127 | 0.520 |
| | Youth *COMT* | 0.029 | 0.823 |
| | <span style="color:red">Youth *DRD2*</span> | <span style="color:red">−0.272</span> | <span style="color:red">0.031</span> |
| | Youth *GR* | −0.077 | 0.548 |
| | Youth *SERT* | −0.113 | 0.378 |
| Total Household Dysfunction | Parent *COMT* | 0.051 | 0.798 |
| | Parent *DRD2* | −0.046 | 0.817 |
| | Parent *GR* | −0.165 | 0.401 |
| | Parent *SERT* | −0.152 | 0.440 |
| | Youth *COMT* | 0.054 | 0.677 |
| | Youth *DRD2* | −0.202 | 0.113 |
| | Youth *GR* | −0.081 | 0.528 |
| | <span style="color:#4a90d9">Youth *SERT*</span> | <span style="color:#4a90d9">−0.232</span> | <span style="color:#4a90d9">0.067</span> |

Note: ACE = adverse childhood experience, *COMT* = catechol-O-methyltransferase, *DRD2* = dopamine receptor, *GR* = glucocorticoid receptor, *SERT* = serotonin transporter, r = Pearson's correlation, *p* = *p* value.

**Table 4.** Results from the correlational analyses between parental ACE measures and epigenetic targets for male offspring.

| Parental ACE Measure | Epigenetic Target | r | p |
|---|---|---|---|
| Total ACEs | <span style="color:red">Parent *COMT*</span> | <span style="color:red">−0.681</span> | <span style="color:red">0.030</span> |
| | Parent *DRD2* | −0.521 | 0.122 |
| | Parent *GR* | −0.354 | 0.316 |
| | Parent *SERT* | −0.240 | 0.504 |
| | Youth *COMT* | 0.052 | 0.824 |
| | Youth *DRD2* | 0.125 | 0.589 |
| | Youth *GR* | 0.059 | 0.798 |
| | Youth *SERT* | −0.094 | 0.686 |
| Total Maltreatment | Parent *COMT* | −0.463 | 0.177 |
| | <span style="color:#4a90d9">Parent *DRD2*</span> | <span style="color:#4a90d9">−0.570</span> | <span style="color:#4a90d9">0.085</span> |
| | <span style="color:#4a90d9">Parent *GR*</span> | <span style="color:#4a90d9">−0.555</span> | <span style="color:#4a90d9">0.096</span> |
| | Parent *SERT* | −0.512 | 0.131 |
| | Youth *COMT* | 0.138 | 0.552 |
| | Youth *DRD2* | −0.100 | 0.666 |
| | Youth *GR* | −0.099 | 0.668 |
| | Youth *SERT* | −0.111 | 0.633 |

**Table 4.** *Cont.*

| Parental ACE Measure | Epigenetic Target | r | p |
|---|---|---|---|
| | Parent *COMT* | −0.737 | 0.015 |
| | Parent *DRD2* | −0.308 | 0.386 |
| | Parent *GR* | −0.010 | 0.979 |
| Total Household Dysfunction | Parent *SERT* | 0.154 | 0.671 |
| | Youth *COMT* | −0.063 | 0.786 |
| | Youth *DRD2* | 0.133 | 0.565 |
| | Youth *GR* | 0.000 | 0.998 |
| | Youth *SERT* | −0.327 | 0.148 |

Note: ACE = adverse childhood experience, *COMT* = catechol-O-methyltransferase, *DRD2* = dopamine receptor, *GR* = glucocorticoid receptor, *SERT* = serotonin transporter, r = Pearson's correlation, *p* = *p* value.

**Table 5.** Results from the correlational analyses between parental ACE measures and pain/psychological outcomes for female offspring.

| Parental ACE Measure | Pain/Psychological Measures | r | p |
|---|---|---|---|
| | Parent PTSD Symptoms | 0.076 | 0.418 |
| | Parent Chronic Pain Status | 0.192 | 0.032 |
| Total ACEs | Youth Pain Interference | 0.026 | 0.774 |
| | Youth PTSD Symptoms | 0.023 | 0.806 |
| | Parent PTSD Symptoms | 0.097 | 0.300 |
| | Parent Chronic Pain Status | 0.218 | 0.015 |
| Total Maltreatment | Youth Pain Interference | −0.009 | 0.918 |
| | Youth PTSD Symptoms | 0.001 | 0.996 |
| | Parent PTSD Symptoms | 0.037 | 0.694 |
| | Parent Chronic Pain Status | 0.121 | 0.182 |
| Total Household Dysfunction | Youth Pain Interference | 0.057 | 0.532 |
| | Youth PTSD Symptoms | 0.042 | 0.657 |

Note: ACE = adverse childhood experience, PTSD = post-traumatic stress disorder, r = Pearson's correlation, *p* = *p* value.

**Table 6.** Results from the correlational analyses between parental ACE measures and pain/psychological outcomes for male offspring.

| Parental ACE Measure | Pain/Psychological Measures | r | p |
|---|---|---|---|
| | Parent PTSD Symptoms | 0.338 | 0.031 |
| | Parent Chronic Pain Status | 0.110 | 0.478 |
| Total ACEs | Youth Pain Interference | 0.153 | 0.327 |
| | Youth PTSD Symptoms | 0.123 | 0.450 |
| | Parent PTSD Symptoms | 0.343 | 0.028 |
| | Parent Chronic Pain Status | 0.118 | 0.447 |
| Total Maltreatment | Youth Pain Interference | 0.101 | 0.521 |
| | Youth PTSD Symptoms | 0.029 | 0.860 |
| | Parent PTSD Symptoms | 0.262 | 0.097 |
| | Parent Chronic Pain Status | 0.080 | 0.608 |
| Total Household Dysfunction | Youth Pain Interference | 0.175 | 0.262 |
| | Youth PTSD Symptoms | 0.200 | 0.216 |

Note: ACE = adverse childhood experience, PTSD = post-traumatic stress disorder, r = Pearson's correlation, *p* = *p* value.

## 3. Discussion

Pediatric chronic pain is a highly prevalent, debilitating, and costly issue that adversely affects not only the individual but also their family and society [4,6,7,46]. The

emergence of chronic pain in youth tends to occur most commonly in late childhood and early adolescence, with longitudinal studies suggesting as much as 64% of these youth will continue to experience chronic pain [47–51] and high rates of mental health problems (e.g., PTSD, anxiety, depression, opioid misuse) into adulthood [5]. Although a variety of factors have been shown to influence the development of chronic pain, genetics and life experiences (i.e., ACEs) have been established as particularly impactful factors in the development and maintenance of chronic pain in youth [2,31]. While little is known about the mechanisms underlying the intergenerational transmission of risk for chronic pain, there is evidence that parental chronic pain status, exposure to ACEs, and trauma symptomology significantly relate to their child's chronic pain outcomes, which suggests that neurobiological and psychological mechanisms likely contribute to the intergenerational transmission of risk for chronic pain [2,18,19,29]. This intergenerational relationship between parental health/experiences and child chronic pain outcomes is also likely modulated by epigenetic changes, given that epigenetic mechanisms respond dynamically to the environment and experiences, producing heritable epigenetic and behavioral changes [34,36,37]. Using an established cohort of youth with chronic pain and their parents, we aimed to determine if the epigenetic changes that modulate the intergenerational transmission of chronic pain could be investigated.

The correlational findings from this study support the hypothesized intergenerational relationships between parent ACEs and parent–child epigenetic changes, chronic pain, and PTSD symptomology. Intriguingly, the genetic biomarkers with the most efficacy were dependent on the gender of the child. More specifically, the results indicate that parental ACE measures significantly correlated with changes in DRD2 expression in female children while significantly correlating with altered COMT expression in the parents of male children. Interestingly, all of these significant correlations were negative, suggesting that as the number of parental ACE exposures increases, the expression of DRD2 in female children and COMT in the parents of male children decreases. The results also indicated that exposure to maltreatment was the driving factor in the relationship between parental ACEs and DRD2 in female children while household dysfunction was the driving factor in the relationship between parental ACEs and COMT in the parents of male children. Given that growing evidence suggests that an individual's physiological response to stress is dependent upon the nature and circumstances of the exposure [52–54], it is not surprising that specific ACEs are associated with different epigenetic changes.

Alterations to the expression levels of these two genes are particularly significant because dysregulation of dopamine functioning within the brain's reward center has been implicated in the sensory and emotional dimensions of chronic pain, as well as its associated comorbidities (i.e., PTSD symptomology, anxiety, and depression) [32,55]. Reductions in DRD2 receptor availability have been associated with reduced pain thresholds [56], and patients with chronic back pain exhibit blunted dopamine release when responding to noxious stimulation [57]. Furthermore, given that DRD2 regulates the synthesis, release, and storage of dopamine [43], this decrease in DRD2 expression in female children may result in blunting of dopamine signaling and could explain the increase in susceptibility to mental health disorders (i.e., PTSD symptomology, anxiety, and depression) in female chronic pain patients [7,58–62]. In addition, mutations that reduce the functionality of dopamine-clearing genes, such as COMT, have been shown to modify pain sensitivity and ultimately affect pain tolerance and emotional states [63].

Although changes in expression of GR and SERT were not significantly correlated with parental ACE variables, they did exhibit trends toward significance and therefore should be included in future investigations. Given the established links between altered expression of the GR and SERT genes and early adversity, enhanced susceptibility to stress, and increased risk for chronic pain, these genes hold promise as objective epigenetic biomarkers [64–68]. While we cannot provide any definitive conclusions with regards to how exactly these epigenetic changes modulate the intergenerational transmission of chronic pain, this brief

report does represent a valuable step towards this objective, by providing evidence for potential epigenetic biomarkers, particularly DRD2 and COMT, for pediatric chronic pain.

The findings that parental ACE measures were significantly and positively correlated with parent chronic pain status in female offspring, and parent PTSD symptoms in male offspring, are consistent with previous research that has found that exposure to adversity in childhood can increase the likelihood of developing chronic pain and/or clinically significant PTSD symptoms in later adolescence and adulthood [14,69–73]. Interestingly, our results suggest that exposure to maltreatment, as opposed to household dysfunction, was driving the correlations between parent total ACEs and their health outcomes. These findings add to an emerging literature on the particular importance of maltreatment-related ACEs for poor health outcomes [16,17]. Current conceptual literature suggests that different types of adversity, such as those related to maltreatment versus household dysfunction, may elicit different physiological and psychological responses and thus differentially impact later health [52,74,75]. However, further research is needed to better understand how various types of adversity may impact the physical and emotional functioning of parents as well as their offspring. While there is a dearth of empirical research on the role of child gender in the intergenerational transmission of chronic pain, previous studies have shown that child gender can moderate the association between parent health (e.g., depression, chronic pain) and child outcomes, with stronger associations typically found for female offspring [76,77]. Our findings suggest that there may be different pathways through which parental ACEs confer risk for poor outcomes in female versus male offspring and underscore the importance of examining child gender in future investigations in this area.

In the current sample, parental ACEs were not significantly related to levels of youth pain interference or youth PTSD symptoms in a sample of youth who all had chronic pain conditions. These findings are similar to a recent study showing the high prevalence of ACEs among parents of youth with chronic pain [78] but no relation to relative levels of pain in these youth who had already developed chronic pain [78]. As previously suggested [78], we hypothesize that parental ACEs *may* confer risk for the onset or development of chronic pain in youth, rather than affect relative levels of pain interference when pain is already chronic. This risk for developing chronic pain may be transmitted directly, through epigenetic mechanisms, or indirectly, through poor physical and mental health in the parent as research has shown that children of parents with (versus without) chronic pain, as well as children of parents with (versus without) elevated anxiety symptoms, are more likely to report chronic pain [18,79]. Further research that prospectively and longitudinally examines the relations among parental ACEs, parental health, epigenetic markers, and child pain is needed in cohorts of at-risk youth who have not yet developed chronic pain at the start of the study (e.g., youth undergoing surgery, sustaining an injury, presenting in primary care with subacute pain).

A final aim of this study was to determine whether epigenetic markers could be examined via saliva rather than blood; we have established this is a viable and less invasive method. The ability to collect saliva instead of blood for epigenetic analyses solves several existing issues in current pediatric chronic pain research. For instance, needle pricks from blood collection can be a confounding factor in pediatric chronic pain research, given that it has the potential to induce pain, fear, and anxiety. By implementing saliva collection instead, this pain and anxiety can be avoided and, in turn, the participants' willingness to contribute to this study measure will likely increase. Saliva is also superior in regard to storage feasibility, since blood requires refrigeration and quick processing while saliva can be stored at room temperature and does not have a processing time requirement. Overall, saliva collection is a viable and arguably preferable method of analyzing epigenetic biomarkers, especially in pediatric chronic pain research, where limiting any unnecessary further pain, fear, and anxiety in children should be of the utmost importance.

There are a few limitations to the current study that need to be considered. First, this study included a cohort of youth with, on average, long-standing and clinically significant

chronic pain that were being assessed and treated in tertiary care, and lacked a control group without chronic pain. Thus, it is possible that more epigenetic changes would have emerged if we had made comparisons between this chronic pain group and a control group without chronic pain. Second, the majority of families in the current study were white and of higher socioeconomic status (i.e., well educated, employed, and high household income). Sociodemographic disparities are integrally related to both trauma and pain [79–82] and the results of this study may not generalize to other populations. Third, since there are likely a large number of factors contributing to gene expression levels, we are unable to account for all possible contributing factors. However, given the significant amount of literature corroborating the relationship between parent pain and/or trauma and child chronic pain [9–17], it is likely that child gene expression levels are influenced by parent pain and/or trauma. Fourth, this study had a relatively small sample size, due to the distinctive nature of the sample we examined (parent and child with chronic pain dyad). Further studies with larger sample sizes are required to increase the statistical power of the current findings. Nevertheless, this study is one of the first to examine how ACEs "get under the skin" to change the epigenomes of parents and children, and confer risk for pediatric chronic pain. By continuing to build on this novel research, we may be able to use noninvasive methods to identify objective biomarkers in parents and their children that predict risk for chronic pain, and ideally mitigate this growing epidemic.

## 4. Materials and Methods

This study is part of a broader research program, entitled the Pain and Mental Health in Youth (PATH) Study, that is examining a myriad of psychological, cognitive, behavioral, neurobiological, and social factors in a clinical sample of youth with chronic pain and their parents. The aims of the current study were distinct from previously published articles that have used data from the PATH Study [19,83–88]. The PATH Study has been approved by the University of Calgary Conjoint Health Research Ethics Board.

### 4.1. Participants

Youth with chronic pain and one of their parents were recruited from tertiary, outpatient chronic pain clinics at a pediatric hospital in Western Canada. Saliva samples were collected from 86 youth (22 males, 64 females) and 41 parents (3 males, 38 females). Youth were eligible to participate if they were 10 to 18 years of age and had ongoing chronic pain (i.e., pain for $\geq 3$ months) that was not associated with an underlying disease (e.g., juvenile idiopathic arthritis, cancer). Youth were not eligible if they were unable to read/speak English, did not have access to the internet, or had any of the following: severe cognitive impairment or developmental disorder, schizophrenia spectrum or other psychotic disorder, or presence of a serious chronic health or life-threatening condition (e.g., cancer). Parents were eligible to participate if they were the legal guardian of the youth, could read/speak English, and had access to the internet.

### 4.2. Procedure

Recruitment procedures have been described in detail elsewhere [83,86,88]. In brief, clinic staff provided the research team with the contact information of families who had recently been referred to, or treated in, the chronic pain clinics. The research team also had access to a list of families who were participating in a clinical outcomes study and consented to be contacted about future research studies. Potential participants were contacted via email or phone with information about the PATH Study. Interested parent–child dyads were screened for eligibility over the phone and an informed consent procedure was conducted wherein the information contained in the consent form was explained, including details of the study and limits to confidentiality, any questions were answered, and agreement to participate was obtained. Written consent and/or assent was also obtained from parents and youth via online forms emailed to dyads after the telephone conversation and hard copy forms provided to dyads at the laboratory visit. All online forms and measures were

administered and completed through Research Electronic Data Capture (REDCap), a secure web-based data collection site [89,90].

Upon enrollment, parents and youth were each sent a battery of self-report measures to complete before the laboratory visit. Specifically, parents completed demographic information and measures of ACEs, chronic pain status, and PTSD symptoms and youth completed measures of pain characteristics and PTSD symptoms. If measures were not completed by the laboratory visit, dyads were provided time to complete the battery during the visit. For the laboratory visit, parents and youth attended the hospital-based laboratory of the research team and provided their saliva sample. Parents and youth each received an honorarium (i.e., $25 CAD gift cards) for their participation in this portion of the PATH Study. Data for the current study were collected between October 2017 and December 2019.

*4.3. Self-Report Measures*

Sociodemographic Information. Parents completed a sociodemographic questionnaire that asked about their own age, gender, race/ethnicity, marital status, education, employment status, and annual household income as well as their child's age, gender, and race/ethnicity. This questionnaire assessed parent and child gender (versus sex) and thus we use the term "gender" in the current study.

Parent ACEs. The Adverse Childhood Experiences (ACE) Questionnaire retrospectively assessed parent exposure to 10 categories of ACEs (i.e., emotional, physical, and sexual abuse; emotional and physical neglect; five types of household dysfunction) in the first 18 years of their life. This 28-item measure was developed for the original ACE Study [12] by adapting items from existing measures of childhood abuse, neglect, and household dysfunction [91–93]. Each category is assessed with one or more items that are rated on dichotomous (yes/no) or five-point Likert-type (0 = "never true" or "never" to 4 = "very often true" or "very often") scales. If at least one item of the category was endorsed, the ACE was coded as present. Total ACE scores were obtained by summing responses to the 10 categories (range: 0–10). Total maltreatment and total household dysfunction scores were obtained by summing responses to the relevant five categories (range: 0–5). Higher scores indicate exposure to more categories of ACEs. This measure has demonstrated good psychometric properties in community and high-risk populations (e.g., low income women, individuals with major depression) [94–97] and studies evaluating its factor structure support the subscales of maltreatment and household dysfunction [95,97]. The measure demonstrated excellent internal consistency (a = 0.95) in the current study.

Parent Chronic Pain Status. Similar to previous research [30,98], and consistent with the current definition of chronic pain [1], parent chronic pain status was assessed with a dichotomous item (yes/no) that asked about the presence of pain for at least three months in a row.

Parent PTSD Symptoms. The PTSD Checklist for DSM-5 (PCL-5) was administered to assess parent PTSD symptoms [99]. This 20-item measure asks respondents to think about the "worst" event they have experienced (i.e., a difficult or stressful event that continues to bothers them) and then rate how much 20 symptoms specific to DSM-5 diagnostic criteria for PTSD have bothered them in the past month on a five-point Likert-type scale (0 = "not at all" to 4 = "extremely"). A total symptom severity score is obtained by summing the ratings for each item (range: 0–80), with higher scores indicating greater PTSD symptoms. The PCL-5 has excellent reliability and validity [100] and has been previously used in research with parents of youth with chronic pain [29,30]. It demonstrated excellent internal consistency (a = 0.93) in the current study.

Youth Pain Characteristics. Youth pain characteristics were used to describe the child sample and assessed with the pain frequency, pain locations, pain duration, and pain intensity items of the widely used Pain Questionnaire [101]. The pain frequency item measures how often the respondent had pain in the past week on a five-point Likert-type scale from "not at all" to "daily". The pain location item asks respondents to select the parts of their body where they experienced the most aches or pains in the past week

from a checklist of six options (e.g., stomach, head, other). The pain duration item asks respondents to indicate how long they have had pain in years and months. The pain intensity item asks respondents to rate the severity of their pain on a validated and reliable 11-point Numerical Rating Scale (0 = "no pain" to 10 = "worst pain possible") [102]. This questionnaire has been used in previous research with youth with chronic pain [29,30,103]. Youth pain interference was assessed with the short form of the Patient Reported Outcomes Measurement Information System (PROMIS)-25 Profile and was considered a primary pain outcome for this child sample. This four-item measure asks respondents to rate the extent to which pain interfered with daily activities such as sleeping in the past week on a five-point Likert-type scale (1 = "never" to 5 = "almost always"). A total score is obtained by summing the ratings for each item and then translating the raw score into a standardized T-score (range: 36.7–74). Higher scores indicate greater pain interference. This measure was developed by the National Institutes of Health using item response theory and has been validated in youth with chronic pain [104]. It demonstrated good internal consistency (a = 0.81) in this study.

Youth PTSD Symptoms. The Child PTSD Symptom Scale for DSM-5 (CPSS-5) was administered to measure youth PTSD symptoms [105]. The CPSS-5 asks respondents to report a scary or upsetting event that happened to them that still bothers them when they think about it. The respondent is then asked to rate how often 20 symptoms specific to DSM-5 diagnostic criteria for PTSD have bothered them in the last month on a five-point Likert-type scale (0 = "not at all" to 4 = "6 or more times a week/almost always"). Total scores were obtained by summing the rating for each item (range: 0–80), with higher scores indicating greater PTSD symptoms. The CPSS-5 has excellent reliability and has demonstrated convergent and discriminant validity [105]. It demonstrated excellent internal consistency in the current study ($\alpha = 0.97$).

### 4.4. mRNA Analysis

A saliva sample (5 ml) was obtained from youths and parents via Oragene DNA OG-500 collection tubes (DNA Genotek Inc., Ottawa, ON) which were then stored at room temperature according to manufacturer protocols. mRNA and DNA were extracted from the saliva samples using the AllPrep DNA/RNA mini kit, in accordance with the manufacturer's protocol (Qiagen, Hilden, Germany). Total RNA concentration and purity were determined by using the Nanodrop 2000, followed by resolving 1 μg of each sample on 1% agarose gel electrophoresis to determine RNA integrity. Two micrograms of purified total RNA was reverse transcribed to cDNA using oligo(dT)$_{20}$ of the Superscript III First-Strand Synthesis Supermix kit (Invitrogen, Carlsbad, CA, USA), as per manufacturer's protocols. Primers were designed in-house using Primer3 (http://bioinfo.ut.ee/primer3/, accessed on 1 February 2021) to span exon–exon junctions based upon the Pubmed "nucleotide mRNA database" and previously described optimization criteria [106], including (a) products size ranging from 150 to 250 bp, maximum = 250 bp; (b) optimal primer size = 22 bp; (c) ~50% GC content; and (d) Tm = 60 °C. Once designed, primers were purchased from IDT (Coralville, IA, USA) and expected amplicon size was confirmed via gel electrophoresis of the PCR product to ensure there was no DNA amplification of the RNA samples. This was followed by a melt curve analysis using SYBR Green FastMix with Rox (Quanta BioSciences, Gaithersburg, MD, USA) to ensure primer specificity. Finally, gradient PCR was performed to determine the optimal annealing temperatures for each primer pair for the target and housekeeping genes. The $2^{-\Delta\Delta Ct}$ method [107] was utilized and normalized against two housekeeping genes (Ywhaz and CycA) to determine changes in mRNA expression [108–110]. See Table 7 for primer sequences, cycling parameters, and PCR efficiencies. Mixtures of 10 ng of cDNA, 0.5 μM of the forward and reverse primers, and 1 X SYBR Green FastMix with Rox were utilized for RT-qPCR analysis using the CFX Connect-Real-Time PCR Detection system (Bio-Rad, Hercules, CA, USA). Standard curves were run for each specific gene, and all samples were run in duplicate.

Four genes were examined as potential biomarkers, given their established links to ACEs and/or chronic pain. The selected genes were COMT, DRD2, GR, and SERT. COMT is involved in the regulation of catecholamine (i.e., dopamine, norepinephrine, epinephrine) and enkephalin (i.e., endogenous opioid peptides) levels [31]. Differing levels of COMT expression and activity have been linked to variations in pain sensitivity, reactivity, and the transition from acute to chronic pain [31,111–115]. Similarly, our group recently demonstrated in a rodent model that early life stress (i.e., maternal separation) elevates COMT expression, suggesting decreased levels of catecholamines and enkephalins are associated with increased anxiety and an altered pain response [42]. DRD2 regulates the synthesis, release, and storage of dopamine, which in turn influences susceptibility to mental health disorders and addiction [43]. Chronic psychological stress and early life stress, such as ACEs, have been shown to alter DRD2 gene expression, thereby promoting the development of depression [43]. GR gene encodes for the receptor that binds glucocorticoids, such as cortisol. Expression of GR has been shown to be downregulated by chronic stress, which is hypothesized to have deleterious consequences for the pain response [44]. Moreover, our group has also found that early life stress in a rodent model increased GR expression, which may have ensued from elevated corticosterone levels in early life [42]. SERT is a monoamine transporter responsible for the reuptake of serotonin and has been implicated in mediating the relationship between life stress and depression [45].

**Table 7.** Primer and cycling information for genes of interest and the respective housekeeping genes (*). The forward primer is denoted with (+) while the reverse primer is denoted with (−).

| Gene Symbol & ID | Gene Name | Primer Sequence | Tm (°C) | PCR Efficiency | Cycling Parameters |
|---|---|---|---|---|---|
| *Comt* (1312) | Catechol-O-Methyltransferase | (+)attcacacctttctgaccaagc (−)ggggacagctctaggtgtagg | 58.0 | 93.60 | |
| *Drd2* (1813) | Dopamine Receptor D2 | (+)gcaatgtatcccttctcacagc (−)aggccaggaatagaaaagg | 55.0 | 94.65 | |
| *GR* (2908) | Glucocorticoid Receptor | (+)tgtatgtgttatctggccatcc (−)tcccatagtttaggcatttgg | 54.0 | 102.63 | 1 cycle 95 °C 3 min; 40 cycles 95 °C 15 s |
| *Sert* (6532) | Serotonin Transporter | (+)tttttcaaagggattggttatgc (−)ttgtcctcggagaagtaattgg | 52.0 | 109.48 | 40 cycles Tm 30 s +Melt Curve |
| *CycA* * (5478) | Cyclophilin A | (+)agcactggggagaaaggatt (−)agccactcagtcttggcagt | 58.0 | 102.20 | |
| *Ywhaz* * (7534) | Tyrosine 3-monooxygenase/tryptophan, 5-monooxygenase activation protein, zeta | (+)ttgagcagaagacggaaggt (−)gaagcattggggatcaagaa | 56.1 | 105.34 | |

### 4.5. Statistical Analysis

All statistical analyses were carried out using SPSS 22.0 for Mac. Descriptive and frequency statistics were conducted to summarize the sociodemographic and pain characteristics of the sample and the key self-report measures. Prior to splitting by youth gender, one-way ANOVAs were conducted to analyze gender differences in gene expression (parent and child) of *COMT*, *DRD2*, *GR*, and *SERT*, with a value of $p < 0.05$ considered significant. After splitting by youth gender, Pearson correlations were conducted. Given that evidence suggests maltreatment is a stronger predictor of chronic pain outcomes than household dysfunction, three parental ACE measures were chosen for these correlations: total ACE score and the two subscales of total maltreatment score and total household dysfunction score. These parental ACE variables were correlated to four parent epigenetic targets (COMT, DRD2, GR, and SERT), four child epigenetic targets (COMT, DRD2, GR, and SERT), two parent pain/psychological measures (parent chronic pain status and parent PTSD symptoms), and two child pain/psychological measures (youth pain interference

and youth PTSD symptoms). As analyses were planned and hypotheses were made a priori, a value of $p < 0.05$ was considered statistically significant for all correlations. Pairwise deletion was used for all analyses; thus, ns in each analysis vary slightly due to missing data on the self-report measures. Missing data ranged from 0% for parent ACEs and parent chronic pain status to 7.3% for parent PTSD symptoms and was considered by Little's MCAR test [116] to be missing at random.

**Author Contributions:** Conceptualization, R.M. and M.N.; Data collection, J.K.B. and M.W.; Data analysis, J.C. and J.K.B.; Writing—Original Draft Preparation, J.C., J.K.B., S.L.O., M.N. and R.M.; Writing—Review and Editing, J.C., J.K.B., S.L.O., M.N. and R.M.; Supervision, R.M. and M.N.; Funding Acquisition, M.N. and R.M. All authors have read and agreed to the published version of the manuscript.

**Funding:** This research was supported by funding awarded to M. Noel from the Vi Riddell Pediatric Pain Initiative (Grant #1036777), Alberta Children's Hospital Foundation and Alberta Children's Hospital Research Institute (Grant #1042861), and the Canadian Institutes of Health Research Strategy for Patient-Oriented Research 'Chronic Pain Network' (Grant #1041605); and to R. Mychasiuk from the Canadian Institutes of Health Research (PJT-153051).

**Institutional Review Board Statement:** This study was conducted according to the guidelines of the Declaration of Helsinki and approved by the Conjoint Health Research Ethics Board of the University of Calgary (REB15-3100; 23 June 2016).

**Informed Consent Statement:** Informed consent from all subjects involved in this study.

**Data Availability Statement:** Data available upon request from corresponding author.

**Conflicts of Interest:** The authors declare no conflict of interest.

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
