# Peer review of "A Pilot Study Investigating the Role of Gender in the Intergenerational Relationships between Gene Expression, Chronic Pain, and Adverse Childhood Experiences in a Clinical Sample of Youth with Chronic Pain"

_2075-4655_

Round 1
Reviewer 1 Report
This is an article review entitled, “Investigating the role of gender in the inter generational relationships between gene expression, chronic pain and adverse childhood experiences any clinical sample abuse with chronic pain.” The article reports to examine the inter-relationship between parental adverse childhood events, parental chronic pain, childhood adverse events and the genetic changes within the offspring of parents and their inter-relationships. This is a novel scientific question to ask, but would be improved by addressing the following concerns.
Abstract
Please clarify the sentence and what is meant by the phrase, “The results indicated that the biomarkers with the most efficacy were dependent on the gender of the child…” please clarify what is meant by “the most efficacy…”
The authors do not clarify if when they are referring to the sample as male and female, do they mean gender or sex? Please clarify.
Introduction
Page 2, The sentence starting “however, we propose that it is also possible that the experiences of a parent with chronic pain and/or trauma produce epigenetic modifications in the child that may put them at heightened risk for the development and maintenance of chronic pain” seems like a very large leap, without resultant evidence to make this assertion. Please address the concern that there may be other possible causes for the genetic modifications in the child that are more proximal than the distal parental experiences.
Page 3, sentence, "Additionally, this study sought to determine the validity of examining epigenetic markers via saliva…” How was the validity tested?
Page 3, The hypotheses are not very clear and are confusing. The authors do not state how they will test the causative nature of the parent’s ACEs “alters the child’s expression levels of the aforementioned genes,…” How was the alteration of the child’s gene expression tested? How does one know that it was the parental ACEs that were causative and not the child's ACEs or the child's pain?
The second hypotheses, is not clearly stated as a directional hypothesis or not tested as such. There is no indication that the prevalence of gene expression alterations were compared between the genders. Males and females gene expressions were not directly compared.
Results
One of the major concerns in the results section include data in the tables demonstrating no relationship between parental ACEs and youth pain status or youth psychological factors. This would seem to be an important 1st step to making the argument that parental ACES were causative of the epigenetic polymorphisms in the children. Since the parental factors do not appear related to the child's pain, it seems unlikely that parental characteristics would then relate to children's genetic polymorphisms.
In order for the authors to make this argument coherently, they need to demonstrate that parental adversity is related to childhood pain. Once that relationship is established, then examining the relationship between parental adverse effects and childhood chronic pain genetic polymorphisms would make sense. A further step will then be to rule out any other possible explanations for genetic polymorphisms in the child that are more proximal to the child than the distal parental adversity.
Methods and Materials
Page 12, it does not appear that the authors controlled for the possibility that the correlation between the parental ACEs and childhood epigenetic morphology in the children was also influenced by parental chronic pain, or the child’s own ACEs or child’s own chronic pain. Therefore, drawing the conclusion that parental ACEs led to the genetic morphology seems unwarranted.
Page 12, statistical analysis
It does not appear that the authors completed a power analysis to determine an adequate sample size to demonstrate adequate power to detect true relationships among the variables. Please provide rationale for not including a power analysis or please include the power analysis.
It appears that many comparisons were made and one wonders if it would be appropriate to use some sort of correction in order to guard against falsely concluding that there were statistically significant relationships when in fact they were due to random chance. Please address.
The author should be commended for attempting to determine the relationship between parental adversity, parental chronic pain and childhood chronic pain characteristics including genetic polymorphisms. This study though has several significant drawbacks which need to be addressed in order to more cogently make the argument that there is indeed a relationship between parental adversity/chronic pain and genetic polymorphisms in their offspring. First, establishing the relationship between parental pain and adversity and child's pain, then ruling out any other possible explanations for the relationship between genetic polymorphisms and the child's pain. Finally, consideration of a control group of parents with chronic pain, ACEs and children without chronic pain that also do not have genetic polymorphisms may help make the case more strongly that parental ACEs are linked to childhood pain.
Author Response
Dear Drs. Metz and Kovalchuk
We are pleased to submit the revisions to our manuscript,Investigating the role of gender in the intergenerational relationships between gene expression, chronic pain, and adverse childhood experiences in a clinical sample of youth with chronic pain to the International Journal of Molecular Sciences; Special Issue on Health and Disease through a Sex and Gender Lens. Please find a detailed point-by-point response to the editor and reviewer comments below.
We have also included a track-changed version of the manuscript and a clean version so that the additions and removals can be easily identified. We would like to thank the reviewer for their comments and suggestions as they have substantially improved the quality of the manuscript. We look forward to any further comments and feedback you may have.
Reviewer Comments:
1.Please clarify the sentence and what is meant by the phrase, “The results indicated that the biomarkers with the most efficacy were dependent on the gender of the child...” please clarify what is meant by “the most efficacy...”Response:We apologize for the confusion with this phrasing. We have now altered the sentence to “The results indicated that the significant biomarkers were dependent upon...”.
2.The authors do not clarify if when they are referring to the sample as male and female, do they mean gender or sex? Please clarify.Response: Our survey asked parents about their child's gender(vs. their biological sex), so we use that terminology in our manuscripts to accurately reflect what we assessed.We have added this clarification to the ‘Self-report measures’sub-section of the Methods section.
3.Page 2, The sentence starting “however, we propose that it is also possible that the experiences of a parent with chronic pain and/or trauma produce epigenetic modifications in the child that may put them at heightened risk for the development and maintenance of chronic pain” seems like a very large leap, without resultant evidence to make this assertion.Please address the concern that there may be other possible causes for the genetic modifications in the child that are more proximal than the distal parental experiences.
Response: Since there are likely a large number of factors contributing to gene expression levels,we were unable to account for all possible contributing factors. However, given the significant amount of literature corroborating the relationship between parent pain and/or trauma and child chronic pain, it is likely that child gene expression levels are influenced by parent pain and/or trauma.However, we have now added this as a limitation to our manuscript.
4.Page 3, sentence, "Additionally, this study sought to determine the validity of examining epigenetic markers via saliva...” How was the validity tested?Response: We apologize for the lack of clarity in this phrasing. We have updated the manuscript to reflect the fact that we have demonstrated saliva has the capacity to be used for mRNA expression analyses and it is not necessary to draw blood from participants.
5.Page 3, The hypotheses are not very clear and are confusing. The authors do not state how they will test the causative nature of the parent’s ACEs “alters the child’s expression levels of the aforementioned genes,...” How was the alteration of the child’s gene expression tested? How does one know that it was the parental ACEs that were causative and not the child's ACEs or the child's pain?Response: Correlations conducted for child pain and child gene expression levels were not significant. Child ACEs were not measured in our questionnaire. Therefore, we correlated child gene expression levels to parental ACEs, which revealed significant differences.We have updated the hypotheses in the introduction to further clarify this point.
6.The second hypotheses, is not clearly stated as a directional hypothesis or not tested as such. There is no indication that the prevalence of gene expression alterations was compared between the genders. Male and female gene expressions were not directly compared.Response: We thank the reviewer for catching this. We had run the analyses demonstrating gender-based differences in gene expression, which is why the analyses were split between males and females. The results from the one-way ANOVAs, for gender differences in gene expression in COMT, DRD2, GR, and SERT have been added. This information has now been included in the Methods and Results sections, with all genes exhibiting significant sex differences in expression,p <.05.
7. One of the major concerns in the results section include data in the tables demonstrating no relationship between parental ACEs and youth pain status or youth psychological factors. This would seem to be an important 1st step to making the argument that parental ACES were causative of the epigenetic polymorphisms in the children. Since the parental factors do not appear related to the child's pain, it seems unlikely that parental characteristics would then relate to children's genetic polymorphisms. In order for the authors to make this argument coherently, they need to demonstrate that parental adversity is related to childhood pain.Once that relationship is established, then examining the relationship between parental adverse effects and childhood chronic pain genetic polymorphisms would make sense.A further step will then be to rule out any other possible explanations for genetic polymorphisms in the child that are more proximal to the child than the distal parental adversity.Response: In the current sample, parental ACEs were not significantly related to levels of youth pain interference or youth PTSD symptoms in a sample of youth who all had chronic pain conditions. These findings are similar to a recent study showing the high prevalence of ACEs among parents of youth with chronic pain [109] but no relation to relative levels of pain in these youth who had already developed chronic pain. As previously suggested [109], we hypothesize that parental ACEs may confer risk for the onset or development of chronic pain in youth,rather than affect relative levels of pain interference when pain is already chronic.This risk for developing chronic pain may be transmitted directly,through epigenetic mechanisms,or indirectly, through poor physical and mental health in the parent as research has shown that children of parents with (versus without) chronic pain, as well as children of parents with (versus without),elevated anxiety symptoms are more likely to report chronic pain [110, 111].
Further research that prospectively and longitudinally examines these relations among parental ACEs, parental health, epigenetic markers, and child pain is needed in cohorts of at-risk youth who have not yet developed chronic pain at the start of the study (e.g., youth undergoing surgery, sustaining an injury, presenting in primary care with sub-acute pain). This has now been added to the Discussion section to address this comment.
8.Page 12, it does not appear that the authors controlled for the possibility that the correlation between the parental ACEs and childhood epigenetic morphology in the children was also influenced by parental chronic pain, or the child’s own ACEs or child’s own chronic pain. Therefore, drawing the conclusion that parental ACEs led to the genetic morphology seems unwarranted.Response: We did conduct correlations between child gene expression levels and parental chronic pain or child chronic pain (child ACEs were not measured in our study), but they did not reveal any significant differences. The significant correlations between parental ACEs and child gene expression levels, however, has allowed us to propose that these alterations in child gene expression levels are partially a result of parental ACEs.
9.It does not appear that the authors completed a power analysis to determine an adequate sample size to demonstrate adequate power to detect true relationships among the variables.Please provide rationale for not including a power analysis or please include the power analysis.Response:Given thatthis study represents a distinct cohort of youth with chronic pain and their parents that are no longer being followed, we are limited to the data that we have collected. We agree that our sample size is small and have therefore decided to change the title to indicate this as a “pilot study”. We have also added this small sample size to the limitations section of the manuscript.
10.It appears that many comparisons were made and one wonders i fit would be appropriate to use some sort of correction in order to guard against falsely concluding that there were statistically significant relationships when in fact they were due to random chance.Please address.Response:We thank the reviewer for their comment. Given that our correlations were based upon a priori hypotheses and were limited to 4 distinct genes (rather than a genome wide association study, RNAseq, or microarray analyses), we feel that the statistical relationships we have established in this pilot study were not due to random chance or false positives, and therefore do not require additional correction such as false discovery rates.
11.The author should be commended for attempting to determine the relationship between parental adversity, parental chronic pain and childhood chronic pain characteristics including genetic polymorphisms.This study though has several significant drawbacks which need to be addressed in order to more cogently make the argument that there is indeed a relationship between parental adversity/chronic pain and genetic polymorphisms in their offspring.First, establishing the relationship between parental pain and adversity and child's pain, then ruling out any other possible explanations for the relationship between genetic polymorphisms and the child's pain. Finally, consideration of a control group of parents with chronic pain, ACEs and children without chronic pain that also do not have genetic polymorphisms may help make the case more strongly that parental ACEs are linked to childhood pain.Response:As noted in the response to comment 3 above, the inability to account for all other contributing factors to gene expression changes has been added as a limitation. In the Discussion section, we have noted the lack of a control group without chronic pain as a limitation to our study. We hope that this pilot study leads to additional research in this area to further elucidate these relations.
Sincerely,
Richelle Mychasiuk
Department of Neuroscience
Central Clinical School
Monash University
6thFloor, 99 Commercial Road
Melbourne VIC, 3004
Richelle.mychasiuk@monash.edu

Reviewer 2 Report
I am attaching my primer analysis with their table and clearly there is a problem that needs sorting.

Round 2
Reviewer 1 Report
This is a re-review of the manuscript entitled, "A pilot study investigating the role of gender in the inter generational relationships between gene expression, chronic pain, and childhood adverse experiences in a clinical sample of youth with chronic pain". it is purporting to examine the relationship between parental chronic pain and childhood adverse events in the expression of childhood chronic painful conditions.
The authors have done a very nice job addressing the concerns raised in the 1st review. As it is currently written there are no new concerns.
With respect to multiple comparisons, a conservative estimate for the number of comparisons is near 50 ( 3 parental ACE conditions X 2 genders X 4 gene expressions). One does begin to wonder whether or not any of these comparisons might be due to chance or due to multiple comparisons. Additionally, the other comparisons that were done between parental ACE measures and pain and psychological characteristics of the child ( 3 ACE measures X 2 genders X 4 pain and psychological characteristics ).
Overall, the authors have done a very nice job in re-writing the manuscript to address concerns raised by the previous review and editor. The paper provides valuable information about relationships between adverse child events and chronic pain status intergenerationally.
Reviewer 2 Report
None
This manuscript is a resubmission of an earlier submission. The following is a list of the peer review reports and author responses from that submission.